# The Ultrastructure of Olfactory Sensilla Across the Antenna of *Monolepta signata* (Oliver)

**DOI:** 10.3390/insects16060573

**Published:** 2025-05-29

**Authors:** Jiyu Cao, Wanjie He, Huiqin Li, Jiangyan Zhu, Xiaoge Li, Jiahui Tian, Mengdie Luo, Jing Chen

**Affiliations:** 1Key Laboratory of Oasis Agricultural Pest Management and Plant Protection Resources Utilization, College of Agriculture, Shihezi University, Shihezi 832000, China; caoisgoodperson@outlook.com (J.C.); lxg1448586789@163.com (X.L.);; 2Yuli Industry Development Service Center of Apocynum Venetum, Yuli 841500, China; 20202012068@stu.shzu.edu.cn; 3Agricultural Science Research Institute, Shihezi 832000, China; qinqindeshijie@163.com; 4Xinjiang Tianye (Group) Co., Ltd., Shihezi 832000, China; yanzhujiang@163.com

**Keywords:** *Monolepta signata*, olfactory sensilla, ultrastructure, antennae

## Abstract

Insects like the *Monolepta signata*, which harms important crops such as maize and cotton, rely on their antennae to smell and find places to lay eggs. We used special microscopes to take a very close look at the antennae of the beetle, discovering different types of sensory structures that help them smell. We found tiny holes on some of these structures, which suggested that they were used to detecting specific smells. This research helps us understand how the beetle find their way to crops and could lead to better ways to pest control, protecting the food supply and reducing crop damage.

## 1. Introduction

*Monolepta signata* (Oliver) (Coleoptera: Chrysomelidae) is a polyphagous pest that feeds on various economic crops including soybean, corn, cotton, potatoes, sunflowers, and so on. It also feeds on weeds such as xanthium, humulus, purslane, abutilon, quinoa, and nightshade [1], categorizing it among widespread feeding pests [2]. This species is primarily distributed in Russia (Siberia), Malaysia, India, Japan, Korea, Vietnam, Singapore, India, Philippines, Indonesia, and other regions of Asia [3,4,5]. The adult beetle prefers to feed on the epidermis of plant leaves, creating nicks and holes. Over time, the affected areas transition from green to brown and eventually dry out, significantly impairing the photosynthesis of plants. Additionally, the adult also feeds on corn silk, grains, and other components during the filling stage, which directly hinders the growth and development of corn. In addition, corn ear rot can occur due to beetle feeding [6,7]. When the damage is serious, it can cause a large area of production reduction, ranging from 15 to 20%, or even no harvest [8]. Beetle larvae live in soil, mainly feed on plant roots, and hinder the normal growth and development of host plants [9]. Currently, chemical agents are being utilized as protective measures aimed at mitigating the adverse effects posed by the beetle.

Insects possess a fundamental sensory structural unit known as sensilla that enables them to perceive and accurately interpret their environment. This sensory capability initiates behaviors such as host-finding oviposition and the selection of suitable habitats for survival. These sensilla can be categorized into several types, including olfactory sensilla, thermo-hygroreceptive sensilla, and mechanical sensilla [10,11,12]. Insects detect chemical signals from their surroundings through olfactory sensilla, which are distributed across surfaces including antennae labial palps, maxillary palps, and other body parts [13,14,15]. The olfactory sensilla, including sensilla trichodea (ST), sensilla basiconica (SB), and sensilla coeloconica (SCo), have functions that can be inferred from the number and arrangement of their pores. Wall-pored ST and tip-pored SCo are predominantly olfactory, with the latter also playing a crucial role in detecting pheromones in species such as *Bombyx mori* and *Helicoverpa armigera* [16,17,18]. The study of the ultrastructure and function of antennal sensilla is fundamental to understanding insect chemistry and behavior. Although previous research on the external morphology of antennal sensilla in *M. signata* has previously described sensilla trichodea, basiconica, chaetica, coeloconica, campaniformia, and Böhm bristles [19], the specifics of their responses to volatiles and the ultrastructure of the antennae remain unreported.

The olfactory receptors of insects are intricately associated with volatile compounds. The sensilla trichodea of *Moth Heliothis* contain sensory neurons that respond to the primary pheromone component, (Z)-1-hexadecenal [20]. In the Asian longhorned beetle (*Anoplophora glabripennis*), olfactory sensory neurons respond to plant-related volatiles such as geraniol and citronellal [21]. Recent studies have demonstrated that female *M. signata* exhibits a strong attraction to β-ionone, Dragosantol, and α-pinene. Conversely, males show a preference for γ-terpene, D-limonene, 1,3-cyclohexadiene, β-caryophyllene oxide, and D-cinene [22]. Notably, both sexes of *M. signata* were found to be attracted to Dragosantol and α-pinene, with their sensitivity to these compounds increasing at a concentration of 10 μL/mL [22,23,24]. Recent studies have identified 114 olfactory genes from the antennae of *M. signata*, which play an important role in the perception of specific volatiles, affecting the olfactory sensitivity and selection differences in male and female adults [25,26]. These complex behavioral responses are closely linked to the functionality of the antennal olfactory sensilla in *M. signata*. In the in-depth study of this insect, it is found that there are still deficiencies in the description of the ultrastructure of its olfactory organs in the existing literature. Therefore, an in-depth study of the ultrastructure of the antennal sensilla of *M. signata* will provide strong support for the molecular mechanism of olfactory and chemical ecology, which is of vital significance for the sustainable pest management of this insect.

In this study, we meticulously examined and characterized the external features, such as types, subtypes, abundance, and distribution, as well as the internal morphology of the sensilla, with a particular focus on the olfactory sensilla located on the antenna of *M. signata*, using scanning electron microscopy (SEM) and transmission electron microscopy (TEM). We aim to infer the physiological functions and roles of these sensilla. This research offers significant reference materials for clarifying the mechanisms underlying antennal recognition and chemosensation. Additionally, it lays a theoretical groundwork for further investigations into olfactory behaviors associated with antennae.

## 2. Materials and Methods

### 2.1. Insects

Adult *M. signata* were collected from corn fields in Dongwan, Shawan, in the Xinjiang Uygur Autonomous Region (85°48′ E, 44°03′ N). Subsequently, they were transported to the laboratory for indoor rearing under controlled conditions: a temperature range of 26–28 °C and relative humidity maintained at 40–50%. The photoperiod was set at 16:8 (L:D), while fresh cotton leaves were provided daily as their food source. Six male and six female adult insects were taken, respectively, for the preparation of antennae samples

### 2.2. Scanning Electron Microscopy (SEM)

To investigate the antennal olfactory sensilla in *M. signata*, male and female antennae were dissected from active adults and fixed for 24 h in 2.5% glutaraldehyde in cacodylate buffer (PBS) to observe them under scanning electron microscopy (SEM). The antennae from each sex were repeatedly rinsed in 0.1 mol/L phosphate buffer and then dehydrated by using ascending concentration gradients with acetone (30%, 50%, 60%, 70%, 80%, 90%, 95%, and 100%) for 15 min per concentration, followed by drying for 12 h. The samples were then mounted on SEM stubs using double-sided adhesive tape, sputter-coated with gold, and examined with a Hitachi SU8010 scanning electron microscope (Hitachi, Tokyo, Japan) at 15 kV.

### 2.3. Transmission Electron Microscopy (TEM)

The antennae were prefixed in 2.5% glutaraldehyde for 24 h. After three 15 min rinses in PBS (0.1 mol/L), post-fixation was performed in 1% osmium tetroxide (OsO_4_) for 2 h; then, they were rinsed in the same buffer three times for 15 min. The antennae were then dissected using microsurgical forceps and scissors and dehydrated in 100% ethyl alcohol through a series of concentration gradients: 30%, 50%, 70%, 90%, and 100%, with each concentration maintained for 15 min. The samples were subsequently immersed in 100% acetone twice, with each immersion lasting 30 min. Next, the material was embedded separately in Epon resin diluted in acetone at a 1:1 (*v*/*v*) ratio for 1 h and a 3:1 (*v*/*v*) ratio for 3 h. The samples were then embedded in pure Epon resin overnight. Following this, the material was transferred to pure Epon resin, which was allowed to polymerize and condense into a resin block at 70 °C. After obtaining the transverse slice of the antennae, ultra-thin sections of about 50 nm were prepared using an ultramicrotome. These ultrathin sections were doubly stained with uranyl acetate and lead citrate, air-dried, and subsequently observed and photographed using a transmission electron microscope (Hitachi HT7700, Hitachi, Hitachinaka, Japan) at 80 kV.

### 2.4. Measurement and Data Analysis

The classification of sensila was based on previous research [16,17,18,22]. We selected five sensilla of each type from the antennae of both males and females, measuring their lengths and basal diameters using Nano Measure (version 1.2.5). Electron microscopy images were processed with Photoshop (version 21.2.4). Male and female antennal sensillum differences in their lengths and basal diameters were analyzed with the independent sample t-test by using SPSS (version 20) for Windows 10.0.

## 3. Results

### 3.1. General Morphology of Antenna

The antennae of *M. signata* are filiform segmented and consist of the scape (Sc), pedicel (Pe) [5], and flagellum (F), which is divided into nine subsegments (F1–F9). The scape and pedicel segments are yellow-brown, while the remaining segments are black (Figure 1). Seven types of sensilla were observed, including sensilla trichodea (types I and II), sensilla chaetica, sensilla basiconica (types I and II), sensilla coeloconica, sensilla campaniformia, sensilla auricillica, and Böhm bristles (Figure 2). The types of sensilla on the antennae of the females and males are same. And there are no statistically significant differences between the length and base diameter of the same sensillum types between the sexes (Figure 3 and Figure 4).

From the perspective of the antennae as a whole. Sensilla trichodea II is located in Flagellum 6–9 subsections; sensilla basiconica I is on the side of the intervertebral fossa of the Flagellum 2–9 subsegments of the antennae; sensilla basiconica II and sensilla chaetica are distributed throughout all segments of the antennae, with smaller quantities on the scape and pedicel. The epidermal protrusion of Sensilla campaniformia is hemispherical, with a conical protrusion at the top (Figure 2D), which can only be observed at the distal segment of the flagellum. Bohm bristles are short, with a relatively rounded tip; have a base deeply embedded in the epidermis; are attached at the internodes of the scape and pedicel, and are distributed laterally (Figure 2E). Sensilla auricillica rolls inward from both sides; has ear-shaped grooves, a blunt and round end, and a small angle with the surface of the antennae; and is nearly parallel (Figure 2B).

### 3.2. Ultrastructure of Antennal Olfactory Sensilla

According to the external and internal morphology, three types of sensory organs in the antennae of *M. signata* are classified as olfactory sensilla.

Sensilla trichodea are the most widely distributed type of sensillum on the antennae of *M. signata*. This type is significantly more numerous on the antennae of males than on those of females and is categorized into types I and II based on their length and morphology.

Sensilla trichodea type I (STI) feature a base that is inserted into a broad socket, exhibiting a longer, hair-like appearance. The entire structure is either straight or curved in a sickle shape, with a part of the sensilla curving proximally. The tip is pointed and slender, while the surface displays prominent longitudinal ridges and mini-pores. Additionally, STI are positioned at an angle of approximately 30° to the antennal. The transverse section through the sensillum resembles a flower with about 12 petals; the epidermis is thick and non-porous, and no nerve dendrites are observed in the lumen. STI are distributed across all segments of the antennae, and the quantity of ST1 in male antennae is significantly higher than that in female antennae. The length of female antennal STI was measured at 57.66 ± 4.15 μm, with a basal diameter of 4.07 ± 0.58 μm; the length of male antennal STI was 60.02 ± 4.66 μm, with a basal diameter of 3.51 ± 0.14 μm (Figure 3 and Figure 4).

Sensilla trichodea II (STII) have a hair-like appearance; are shorter than STI; have a slightly bluntly rounded tip, a smooth surface, and no longitudinal ridges; and are positioned at an approximate angle of 30° to the antennal surface (Figure 2B and Figure 5A). The walls of the sensillum are monolayered and contain four neuronal dendritic branches (Figure 5B,C) with a socketed structure (Figure 5D). STII are primarily distributed in the antennal flagellum. The female STII measure 25.76 ± 2.19 μm in length and 2.48 ± 0.24 μm in basal diameter, while the male STII measure 21.25 ± 2.24 μm in length and 2.03 ± 0.44 μm in basal diameter (Figure 3 and Figure 4).

Sensilla basiconica I (SBI) have elongated conical protuberance and are situated in a peripheral protuberance cavity with a perforated surface, which is constricted at the base and pointed at the tip (Figure 2C and Figure 6A). Sensilla basiconica possess a porous cuticular wall containing approximately 50 internal dendrites and three microtubules in each branch (Figure 6C,D). The average length of female SBI is 13.04 ± 1.65 μm, with a base diameter of 1.59 ± 0.24 μm. In contrast, the average length of male antennal SBI is 10.24 ± 1.33 μm, and the base diameter measures 1.69 ± 0.26 μm (Figure 3 and Figure 4).

For sensilla basiconica II (SBII), the diameters of the upper and lower ends of the sensillum do not differ. The sensillum is almost cylindrical, is situated in a socket on the surface, and is partly curved towards the tip of the antennal as a finger (Figure 2B and Figure 6B). The length of the female SBII is 9.21 ± 0.42 μm with a basal diameter of 1.95 ± 0.04 μm; the length of the male SBII is 9.51 ± 0.99 μm with a basal diameter of 2.01 ± 0.26 μm (Figure 3 and Figure 4).

Sensilla coeloconica (SCo) exhibit an overall shape similar to SBII, but they are half the length of SBII. The sensillum resembles a flower bud and is located in a pit that protrudes from the surface of the antennae. The bottom wall of the sensilla features a vertical groove (Figure 7A). Transverse sections observed under transmission electron microscopy revealed a thin single wall of the sensillum with distinct pores and 14–17 branches within the lymphatic cavity, each containing a single microtubule (Figure 7B–D). The female SCo measure 5.79 ± 0.39 μm in length and 1.71 ± 0.17 μm in basal diameter, while the male SCo measure 5.28 ± 0.24 μm in length and 1.48 ± 0.19 μm in basal diameter (Figure 3 and Figure 4).

## 4. Discussion

In this study, we observed the external morphology and internal structure of the antennal sensilla of *Monolepta signata* using scanning electron microscopy (SEM) and transmission electron microscopy (TEM). We identified seven types of sensilla on the antennal of *M. signata*, including sensilla trichodea (types I and II), sensilla chaetica, sensilla basiconica (types I and II), sensilla coeloconica, sensilla campaniformia, auricular sensilla, and Böhm bristles. Among these, sensilla trichodea, sensilla basiconica, and sensilla coeloconica have been shown to possess olfactory functions. In addition to the various types of sensilla, the surface of the antennae of *M. signata* also features pore-like structures and unique scale-like structures. The morphology of sensilla campaniformia observed in this study closely resembles that of grooved peg sensilla found on the antennae of *Phyllotreta striolata* [27]. It is proposed that these may represent the same type of sensilla, differing only in nomenclature. Additionally, the cuticular pores of the antennae are predominantly located near sensilla trichodea, which are commonly found in Coleoptera, particularly among weevil beetles. This type of sensilla is also referred to as luminal sensilla in *Calosoma maximoviczi* [28].

Sensilla trichodea are one of the various types of sensilla found in the antennae of *M. signata* and are classified into type I and type II based on their morphology. Small pores characterize these sensilla, and they consist of the epidermis, pore, foramen, pore channel, lymphatic fluid, dendrites, and dendritic membrane, which contain one or more olfactory receptor neurons [29]. Sensilla trichodea on the antennae of various lepidopteran females, such as *Trichoplusia ni*, *Helicoverpa armigera*, and *Helicoverpa zea*, and others have been shown to recognize the sex pheromones of conspecifics and influence behaviors such as aggregation and egg-laying [30]. It is hypothesized that moth sex pheromone receptors are expressed in the olfactory neurons of sensilla trichodea. Some research has utilized fluorescent labeling techniques to identify the odor receptor ApolOR1 and its associated pheromone-binding proteins (PBPs), all of which are expressed in sensilla trichodea [31]. In Coleoptera, sensilla trichodea on the antennae of *Melanotus villosus* have also been shown to function as sensory pheromone receptors [32], while sensilla trichodea in *Tribolium castaneum* play a role in odor recognition, aiding in locating host plant and finding mates [33].

Sensilla basiconica are characterized by their thin, delicate walls and distinct pores, which can be observed under a transmission electron microscope (TEM). Porous structures are clearly visible, traversing the internal cavities of the epidermal sensilla. The internal architecture of sensilla basiconica closely resembles that of sensilla trichodea, which house multiple neurons. However, while sensilla trichodea are primarily involved in pheromone detection, sensilla basiconica are chiefly responsive to common environmental odors and exhibit sensitivity to volatiles from host plants [34]. PBPs are localized within sensilla trichodea, whereas general odorant binding proteins (GOBPs) are predominantly found in sensilla basiconica [35]. In *Agrilus planipennis*, AplaOBP1 is primarily expressed in sensilla basiconica I and III, demonstrating strong binding properties with five terpene volatiles of the host plant. Meanwhile, AplaOBP2 and AplaOBP3 are expressed in the hemolymph of sensilla basiconica I, binding to aldehydes and ketones. This evidence underscores the pivotal role of sensilla basiconica in the recognition of volatiles in the host plant.

Sensilla coelocinica exhibit a distinctive conical morphology reminiscent of a long flower bud. Similar to sensilla basiconica, they function as olfactory sensory organs capable of detecting chemical cues. Double fluorescence hybridization experiments revealed that the ionotropic receptors SgreIR8a and SgreIR25a are co-expressed in the cells of sensilla coelocinica in *Schistocerca gregaria* [22]. Furthermore, research using in situ hybridization demonstrated that the ionotropic receptors HarmIR8a and HarmIR25a are predominantly expressed in the sensilla coelocinica of *Helicoverpa armigera*, with no expression detected in other types of sensilla. Single-sensory organ recordings also indicated that sensilla coelocinica in *H. armigera* can detect acids, ammonia, aldehydes, and volatile esters [36]. Additionally, studies on *Bombyx mori* antennae showed that sensilla coelocinica respond to compounds such as 3-hexen-1-ol and 2-hexenal, which attract *B. mori* larvae [37]. These findings confirm that sensilla coelocinica play a crucial role in recognizing host plant volatiles and guiding insects to suitable oviposition sites.

Three types of sensilla on the antennal flagellum of *M. signata* were observed using SEM and TEM techniques, and their differences between adult females and males were analyzed. Studies have shown that olfactory receptor neurons (ORNs) bind to specific volatile molecules. ORNs in sensilla trichodea and basiconica express odorant receptors (ORs), while those in sensilla coeloconica express ionotropic receptors (IRs), indicating that these three types of sensilla primarily function within distinct sensory subsystems [38,39,40]. Previous studies have identified 46 kinds of ORs, 15 kinds of IRs, and 23 kinds of gustatory receptors (GRs) in the antennal of *M. signata*. Future studies can further analyze the genes contained in each receptor and their corresponding receptors to more fully reveal the sensory and behavioral regulatory mechanisms of *M. signata* [26,41]. Based on these structural characteristics and the corroborating literature, the possibility of further connection that sensilla trichodea, basiconica, and coeloconica house chemoreceptors that respond to volatile stimuli is valid. Further validation will require more sophisticated approaches, such as electrophysiological recordings of responses to selectively test volatile organic compounds for each type of sensillum. Therefore, future research is essential to fully elucidate the connections between the sensilla and the behavioral mechanisms of *M. signata* across different organs. This will facilitate the revelation of the coevolutionary relationship between sensilla and host plants, the screening of plant-resistant varieties, and the development of efficient monitoring, interference, and trapping techniques [42], thereby achieving precise pest control and ecological protection.

## Figures and Tables

**Figure 1 insects-16-00573-f001:**
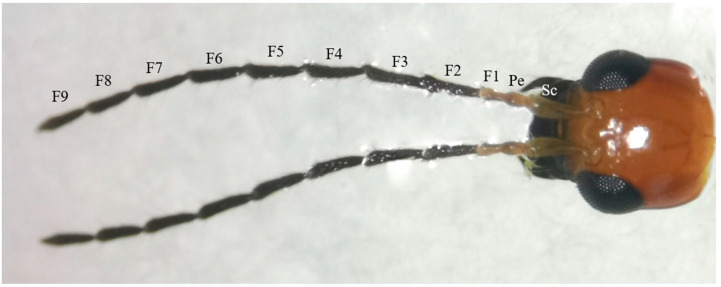
Morphology of the head of *M. signata.* F1–F9: flagellomere 1–9; Pe: pedicel; Sc: scape.

**Figure 2 insects-16-00573-f002:**
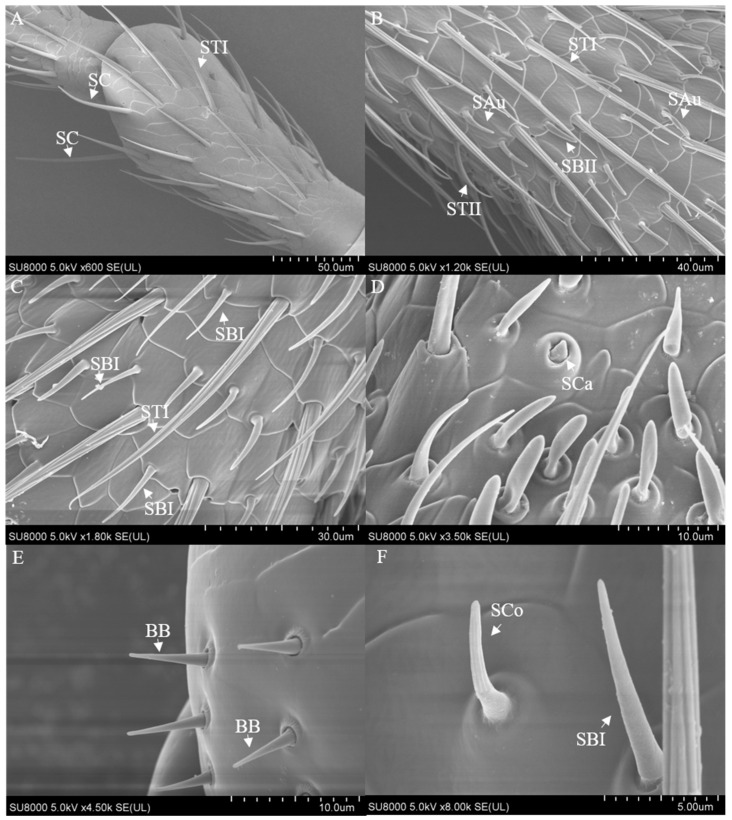
The sensilla on the antennae of *M. signata* (SEM). (**A**–**D**,**F**) Sensilla on the flagellum; (**E**) Sensilla on the pedicel. BB: Böhm bristles; SAu: sensilla auricillica; SB I and II, sensilla basiconca I and II; SC, sensilla chaetica; SCa, sensilla campaniformia; SCo, sensilla coeloconica; ST I and II: sensilla trichodea I and II.

**Figure 3 insects-16-00573-f003:**
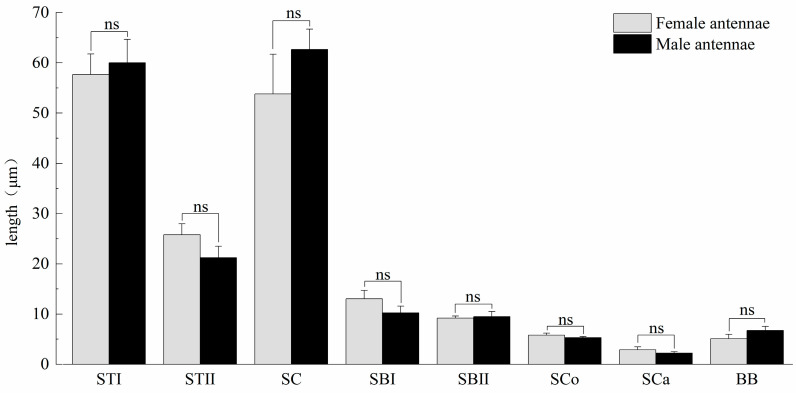
Length of antennal sensilla of *M. signata.* BB, Böhm bristles; SBI, sensilla basiconica I; SBII, sensilla basiconica II; SC, sensilla chaetica; SCa, sensilla auricillica; SCo, sensilla Coeloconica; STI, sensilla trichodea I; ST II, sensilla trichodea II. ns means no significant difference (*p* > 0.05).

**Figure 4 insects-16-00573-f004:**
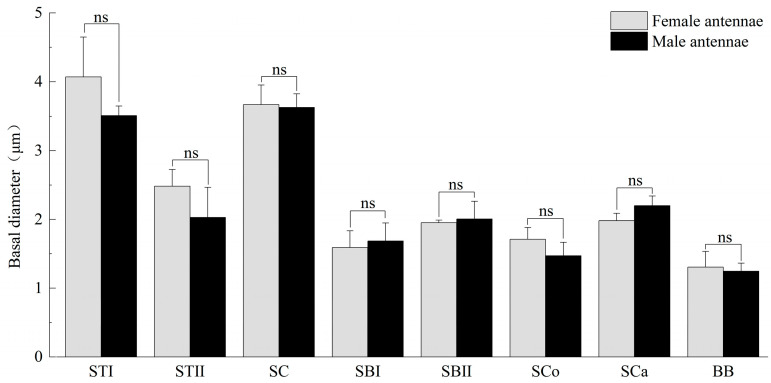
Basal diameter of antennal sensilla of *M. signata.* STI, sensilla trichodea I; ST II, sensilla trichodea II; SC, sensilla chaetica; SBI, sensilla basiconica I; SBII, sensilla basiconica II; SCo, sensilla Coeloconica; SCa, sensilla auricillica; BB, Böhm bristles. ns means no significant difference (*p* > 0.05).

**Figure 5 insects-16-00573-f005:**
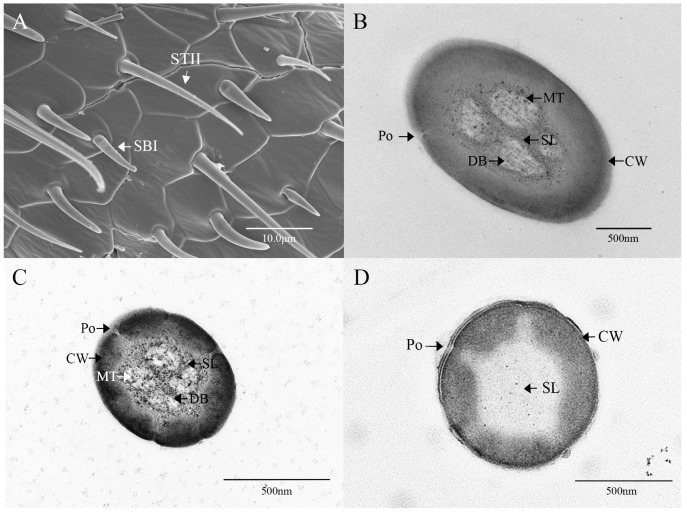
Ultrastructure of sensilla trichodea II on the antennae of *M. signata.* (**A**) External morphology of sensilla trichodea II; (**B**–**D**) cross-section of sensilla trichodea II. CW: cuticular wall; SL: lymph space; Po: pore; DB: dendritic branches; MT: microtubules.

**Figure 6 insects-16-00573-f006:**
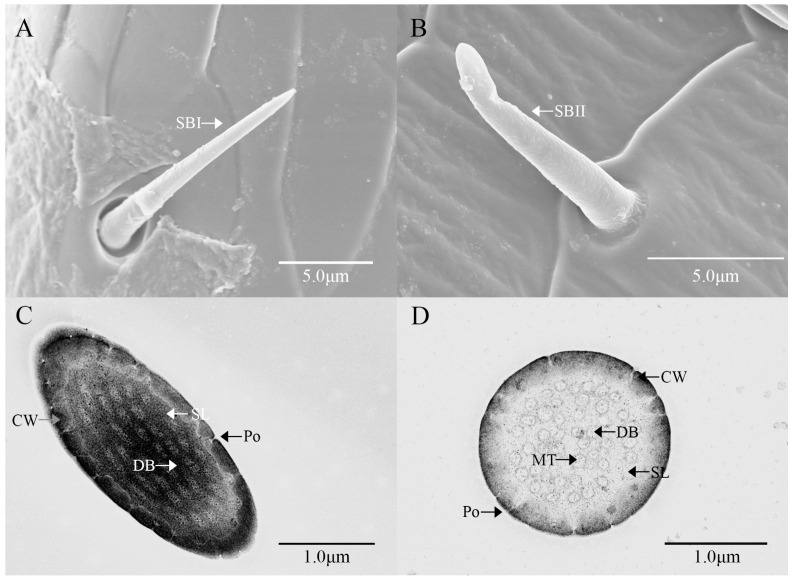
Ultrastructure of sensilla basiconca on the antennae of *M. signata.* (**A**,**B**) External morphology of sensilla basiconca I and II; (**C**) oblique section of sensilla basiconca I; (**D**) cross-section of sensilla basiconca I. CW: cuticular wall; DB: dendritic branches; MT: microtubules; Po: pore; SL: lymph space.

**Figure 7 insects-16-00573-f007:**
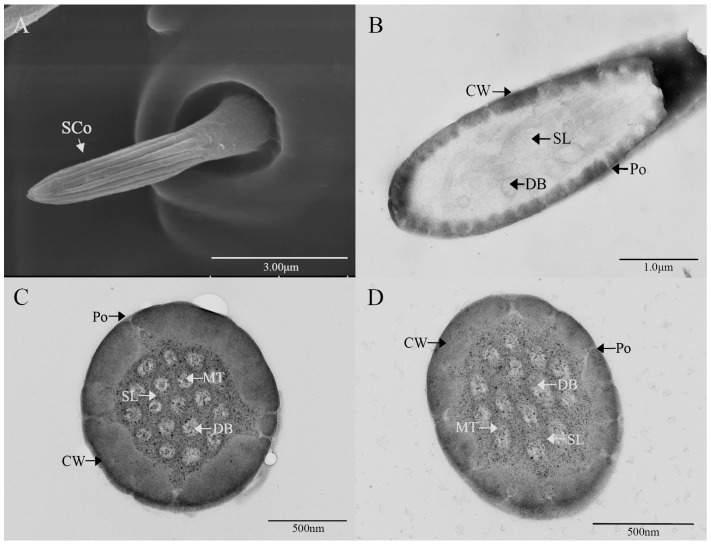
Ultrastructure of sensilla coeloconica on antennae of *M. signata.* (**A**) External morphology of sensilla coeloconica; (**B**) oblique section of sensilla coeloconica; (**C**,**D**) cross-section of sensilla basiconca. CW: cuticular wall; SL: lymph space; Po: pore; DB: dendritic branches; MT: microtubules.

## Data Availability

The original contributions presented in the study are included in the article. Further inquiries can be directed to the corresponding authors.

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
