# Peer review of "The Ultrastructure of Olfactory Sensilla Across the Antenna of Monolepta signata (Oliver)"

_insects, 2025, doi:10.3390/insects16060573_

Round 1

Reviewer 1 Report

Comments and Suggestions for Authors

The ultrastructure of olfactory sensilla across antenna of Monolepta signata (Oliver)

Jiyu Cao, Wanjie He, Huiqin Li, Jiangyan Zhu,  Xiaoge Li, Jiahui Tian, Mengdie Luo, and Jing Chen

This paper describes the various antennal sensillum types found in the grain crop pest, Monolepta signata. The authors have described three olfactory sensillum types and four non-olfactory sensillum types.

Seven types of sensilla were observed, including sensilla trichodea (types I and II), sensilla  chaetica, sensilla basiconica (types I and II), sensilla coeloconica, sensilla campaniformia, sensilla auricillica and Böhm bristles.

Specific critique comments

Lines:

143- capitalize b  of Böhm bristles

166- three of the types of sensory organs….

168- Sensilla trichodea are the most…

169- This type is significantly more…

171- use socket, if appropriate for this sensillum type, instead of fossa; should include approximate length of sensillum here. Where is the reference to the specific figure? Fig. 2???

174- how are you defining micropores versus pores?

175- what are “V”-shaped lines

176- what are “antennal horns”? Also, “transverse section through the sensillum…”

177- no dendrites observed? Through the entire sensillum? None at the base near the insertion of the sensillum into the socket? Also, delete “nerve”.

184- what do you mean by monolayered?

185- do you mean pore at the tip?

190- what are “V”-shaped lines?

192- how do you know that it is a “thin, porous epidermal wall”? Measurements? - Also, the term “epidermal” should be replaced with “cuticular”

198, 199- remove “much”. Start a new sentence after “differ”. The sensillum is almost cylindrical, etc. Could say “socket” rather than “depression that is slightly raised all around”

200- 199- use of “micropores” again (see above)

203- typo for “Coeloconica”

204- do not use comparison to a specific flower, as readers may not know what this is

205, 206- use of vertical stipe? and micropores (see above)

206- what do you mean by a monolayer? Monolayer of what? What are “transsections”? Do you mean “transverse sections”

207- see comments about monolayer

208- cannot see the “single microtubule” due to poor quality of sections. Also, images in general for such transverse sections should be of higher magnification.

230, 231- typo for coeloconica

232, 233, 234- fine to say that these sensillum types have “been shown to possess olfaction functions”, but pores are not visible from any of your SEM images. Those in TEM images have been either compromised with staining artifacts, poor focus, or magnification issues. “Scale-like structures” were not clearly pointed out,

237- use cuticular pores, rather than epidermal. Again, cannot see the pores clearly.

242-245- “small pores”- where are they? These small pores do NOT consist of epidermis, pore, etc. . This is an incorrect statement.

245-254- Again, being able to visualize pores in your figures to make such statements is critical

256, 257- “clearly visible”- see comments, above

269- leave out flower name (see above)- can say “flower-like appearance”

276, 277- SEM images failed to show pores present in order to detect such compounds

290- spaces required between “it” and “[26, 41]

291, 292- “was” confirmed in that study? The word, “is” , was used making it sounds like it was confirmed in the present study. You cannot draw such conclusions.

General Critique of Paper

-The authors have failed to provide the appropriate high magnification of the various sensilla. One cannot visualize most features described. This makes it difficult and flawed to report some of the statements reported in the Discussion section. If the authors have failed to show the necessary magnifications or quality of specific structures in their sections or images, they cannot make the conclusions that they are reporting. For example, the results from the image magnifications provided (especially in the SEMs) fail to show pores. Such structures cannot be resolved in these images. Such a comment pertains to e.g., sensilla shown in Figs. 2A-D, F, 5A where they describe STI, STII, SBI, SBII, etc.

What are the names of the other sensillum types, other than STII, shown in Fig. 5A?

You did not further describe the sensilla campaniformia, auricillica, or Böhm’s bristles, if you point these sensilla out in your figures? Even though you may not think that they bear an olfactory in function, you should still describe them, at least briefly. If your images were generally of higher resolution quality, pores may actually be visible on some sensilla, like the coeloconica, auricillica, trichodea, and basiconica sensilla where you either failed to describe them or did show TEMs but could not back this up with SEM images. Such, sensillum types have been reported in the literature to have an olfactory function. Perhaps, you are missing such structures due to resolution and magnification issues.

-There is much artifactual stain precipitate in Figs. 5B-D, 7C, D that makes it very difficult to see the dendrites and pores in general and/or in comparison to the surrounding sensillum lymph fluid. Fig. 5 C,D are of poor quality, making it difficult to see pores, dendrites, sensillum lymph clearly, as are Figs. 6C, 7 C, D.

-When the authors mention e.g. “V-shaped lines”, as well as “micropores” in the various sensillum descriptions, they have not been referred to or pointed out in the figure legends or shown in the microscopic images.

-What are the other sensilla shown in the bottom right section of Fig. 2D? Are these also SAu sensilla?

Figs. 6A, B- images should be more in focus. You do not point out any features of the sensilla here e.g., SBI, socketed, SBII, unsocketed, etc.

Figs. 7A, B- Pores appear to be present in Fig. B-D. Why do we not see them in Fig. 7A.? The sensillum in 7A needs to be shown in higher magnification so that you can resolve pores to match the commentary about the same sensillum type described in your TEM sections. Such a problem also applies to other sensillum types.

It is unclear as to which sensilla are socketed versus socketed.

Reviewer 2 Report

Comments and Suggestions for Authors

This paper describes the surface and ultrastructure of the antennal sensilla of a species of beetle of agricultural interest as a crop pest. Attention is given predominantly to the olfactory sensilla due to potential relevance for the development of more effective chemical lures for this species.

The methods are reasonably well described however there are some key details missing including the number of male and female beetles whose antennae were studied and whether only one or both antennae from each individual was examined.

In the results, there is some indication of "differences" in certain metrics despite their being no statistically significance to this pattern; this is a cause of ambiguity, and the authors should refrain from describing differences that aren't significant since they are employing probabilistic statistics. Additionally, there is discussion of differences in the numbers of some types of sensilla between females and males, yet there is no figure or table presenting any such data nor is a t value or p value provided to substantiate these claims of a significant difference.

I also have some concerns with some of the sensilla typing as, in some of the figures, the same sensilla type is attributed to sensilla that have different external morphology from each other. Additionally, do you have the ultrastructural information for the sensilla types that you did not identify as being olfactory? If so, putting that in just one figure and having a paragraph summarising the findings (aporous suggesting no olfactory function, etc) would be useful even if included as supplementary information rather than as part of the main manuscript.

I have made specific annotations on a copy of the manuscript (attached) regarding these issues (and others) in both the methods and results section. There is considerable detail lacking from elements of both of these sections of the manuscript.

Broadly, I think this manuscript has missed an opportunity to gain some more informative data from the SEM images likely already obtained. For example, if the number of each type of sensilla on a given region of antenna has been calculated then so could the sensilla abundance or density, i.e. the number of sensilla per unit area. Sensilla density has been shown to have a positive relationship with electrophysiological and behavioural responses to olfactory stimuli (Spaethe et al 2007 in Naturwissenschaften, Gill et al. 2013 in The American Naturalist, Elgar et al. 2018 in The Yale Journal of Biology and Medicine) and so would provide greater insights as to whether one sex may be more or less sensitive to different types of olfactory information if there are sex differences in the abundance of a particular type of olfactory sensilla.

Finally, regarding the conclusions in the discussion. I really like that you've identified clear areas for future research including the need for electrophysiological testing of the response of each type of olfactory sensilla to different kinds of odours/chemicals. However, there is no strong link back to the concept of pest control for this species which is a considerable part of the rationale on which your study was established in the introduction section of the article. I suggest making this link so as to strengthen the story your paper is telling.

I hope this feedback is received in the constructive spirit in which it was intended. 

Comments on the Quality of English Language

I think the English is of an impressive quality for what seems likely to be authors for whom English is a second language. In the introduction especially, there are some sections in which the expression is quite awkward or terse and this impacts how easily the article can be read.

Reviewer 3 Report

Comments and Suggestions for Authors

Dear authors, your manuscript submitted accomplished merit and originality, which, from my point of view, is acceptable; however, I realized that some points must be addressed before going ahead:

1. The instruction still needs to include more comparative studies, e.g. your research VS other similar work, especially btw lines 69 to 80. Take care with these statements.

2. Section 2.4, needs a broader explanation. Please expand the description of your data analysis

3. Include in Materials and Methods, an abbreviation list of all used acronyms throughout the manuscript.

4. Discussion could be improved by the inclusion of other comparative studies similar to yours; the discussion needs to cover overall data found versus comparisons to other similar taxa.

Comments on the Quality of English Language

needs to focus on discussion, some finder errors are found, double check

Round 2

Reviewer 2 Report

Comments and Suggestions for Authors

The authors have responded to most of the comments from the original round of review. It would be good to see the authors clearly mention the need for future studies to examine the sensilla density on the antennae with an explanation of why this has greater behavioural relevance than sensilla number but otherwise I think this manuscript is acceptable for publication.